# Optimized Production of a Redox Metabolite (pyocyanin) by *Pseudomonas aeruginosa* NEJ01R Using a Maize By-Product

**DOI:** 10.3390/microorganisms8101559

**Published:** 2020-10-10

**Authors:** Francisco Javier Bacame-Valenzuela, Jesús Alberto Pérez-Garcia, Mayra Leticia Figueroa-Magallón, Fabricio Espejel-Ayala, Luis Antonio Ortiz-Frade, Yolanda Reyes-Vidal

**Affiliations:** 1Centro de Investigación y Desarrollo Tecnológico en Electroquímica (CIDETEQ), Parque Tecnológico Querétaro s/n, Sanfandila, Pedro Escobedo, Querétaro C.P. 76703, Mexico; fbacame@cideteq.mx (F.J.B.-V.); jgarcia@cideteq.mx (J.A.P.-G.); mfigueroa@cideteq.mx (M.L.F.-M.); fespejel@cideteq.mx (F.E.-A.); lortiz@cideteq.mx (L.A.O.-F.); 2Consejo Nacional de Ciencia y Tecnología (CONACYT)—Centro de Investigación y Desarrollo Tecnológico en Electroquímica (CIDETEQ), Parque Tecnológico Querétaro s/n, Sanfandila, Pedro Escobedo, Querétaro C.P. 76703, Mexico

**Keywords:** pyocyanin, maize industry wastewater, *Pseudomonas aeruginosa* NEJ01R, redox metabolite, optimization, response surface analysis, bioproduction, valorization, electrochemical analysis

## Abstract

*Pseudomonas aeruginosa* metabolizes pyocyanin, a redox molecule related to diverse biological activities. Culture conditions for the production of pyocyanin in a defined medium were optimized using a statistical design and response surface methodology. The obtained conditions were replicated using as substrate an alkaline residual liquid of cooked maize and its by-products. The untreated effluent (raw nejayote, RN) was processed to obtain a fraction without insoluble solids (clarified fraction, CL), then separated by a 30 kDa membrane where two fractions, namely, retentate (RE) and filtered (FI), were obtained. Optimal conditions in the defined medium were 29.6 °C, 223.7 rpm and pH = 6.92, which produced 2.21 μg mL^−1^ of pyocyanin, and by using the wastewater, it was possible to obtain 3.25 μg mL^−1^ of pyocyanin in the retentate fraction at 40 h. The retentate fraction presented the highest concentration of total solids related to the maximum concentration of pyocyanin (PYO) obtained. The pyocyanin redox behavior was analyzed using electrochemical techniques. In this way, valorization of lime-cooked maize wastewater (nejayote) used as a substrate was demonstrated in the production of a value-added compound, such as pyocyanin, a redox metabolite of *Pseudomonas aeruginosa* NEJ01R.

## 1. Introduction

Microorganisms can produce a large variety of extracellular compounds with different biological activities. Currently, there are more than 50,000 natural products used as drugs that are derived from microorganisms, mostly isolated from soil samples [1]. The genus *Pseudomonas*, Gram-negative aerobic Gammaproteobacteria belonging to the family Pseudomonadaceae, is known to produce a series of extracellular redox metabolites of the phenazine group. Species of the genus *Pseudomonas* are characterized by their metabolic versatility, giving them great relevance in environmental recycling processes, degrading a wide range of simple and complex organic compounds [2,3,4]. It has been demonstrated that *Pseudomonas aeruginosa* produces 5-methylphenazine-1-one or pyocyanin (PYO), a water-soluble blue-green pigment with a redox potential similar to menaquinone [5].

PYO is a biomolecule that participates in the quorum-sensing process of *P. aeruginosa*. This bacterial mechanism is used to coordinate population density, as well as for the regulation of gene expression and biofilm formation [6,7]. *Pseudomonas* integrates a bacterial strategy through PYO molecules for the inhibition of fungal phytopathogens of agricultural crops and cytostatic activity against cancer cell lines [8]. The antagonistic effect of the compounds derived from phenazines is associated with their redox behavior. Therefore, it is believed that many capabilities of PYO and other phenazine derivatives, in a variety of eukaryotic hosts and bacteria, are the result of oxidative activity or inactivation of important proteins in response to oxidative stress [9].

The redox process for PYO was studied through cyclic and square wave voltammetries. PYO and other redox metabolites included in the secretome of *P. aeruginosa* could be used as electrochemical biomarkers to detect its presence in a bacterial culture supernatant [10]. Furthermore, *P. aeruginosa* is considered an electrogenic microorganism and is used in bioelectrochemical systems for electrical energy production, carried out by direct electron transfer (through cytochromes in outer membrane or bacteria pili) and indirect electron transfer (through secondary metabolites excreted by bacteria). In this case, PYO of *P. aeruginosa* is the extracellular metabolite that facilitates electronic transfer between microorganism and electrode due to its reversible redox properties [11]. However, one disadvantage of PYO is its market cost since 5 mg (purity > 98%, HPLC grade) has a cost of 60–97 USD according to different suppliers [12,13,14,15]. Therefore, optimized processes must be developed that allow the metabolite to be obtained at high concentrations through operations less harmful to the environment, such as green chemistry, to extend the use of PYO.

An attractive approach is to use residues as substrates in bioprocesses to obtain value-added molecules. In this way, PYO production by submerged fermentation of *P. aeruginosa* has been carried out in glycerol (residues of biodiesel production) and a defined medium supplemented with a broad variety of raw materials [5,16,17]. One of the options to achieve this goal is the use of wastewaters as a carbon source, which could be considered toxic for the environment, but the use in culture media is an alternative that reduces its polluting and harmful potential, as well as its valorization.

Currently, among the agro-food-processing industrial effluents considered as environmental pollutants are by-products generated in the maize (*Zea mays*) cooking process. This traditional process is known as “nixtamalization” and includes boiling corn grains in a saturated solution of calcium hydroxide (0.5–2%) at 90 °C for 40 min; after this process, the corn grains are steeped for 12 h, then drained, and the resulting liquid is commonly known as nejayote [18,19]. Cooked maize grains (nixtamal) are used for the production of masa, tortillas and derived products in Mexico, Southern United States, Central and South America, Asia and parts of Europe. In Mexico, it is estimated that 14.4 million m^3^ of nejayote is produced annually [20]. This wastewater is considered as a pollutant due to its contents extracted during the process: 0.5–14.5% of the corn’s weight goes to the effluent, which has a pH of 12 [18]. Various compounds and some phytochemicals associated with the cell wall of corn grains are released by alkaline hydrolysis and are related to parameters such as total organic carbon (TOC; 2700–59,000 mg L^−1^), biochemical oxygen demand (BOD; 2.69 mg O_2_ L^−1^) and chemical oxygen demand (COD; 7500–40,000 mg L^−1^) [21,22,23,24]. Some of these parameters also indicate that the effluent has a high content of organic material (reducing and total sugars), which together with other compounds (protein, fiber, fat, calcium, arabinoxylans and polyphenols) define its content of nutrients (carbon source) and inorganic compounds, potentially applicable for biotechnological applications [25,26,27,28,29].

Likewise, this wastewater can be used as a raw substrate by microorganisms to obtain bioproducts. *Aspergillus oryzae* and different species of *Lactobacillus* produce a protease and bacteriocins, respectively, using nejayote as a substrate [30,31]. In other studies, two native isolated bacteria (*Bacillus flexus*) were used to biosynthesize amylases, xylanases, proteases and phenolic acid esterase [32]. Recently, *Bacillus megaterium* (a native nejayote strain) was able to transform ferulic acid present in the effluent into 4-vinylguaiacol [33]. Moreover, nejayote in combination with other residual effluents such as vinasse (a waste from tequila production) and swine wastewater allow for obtaining bioenergy or can used for the growth of microalgae [24,34]. Thus, it is clear that components of nejayote can be exploited for the production of different biotechnological products. Therefore, the objective of this work was to valorize lime-cooked maize wastewater as a culture medium to produce a redox metabolite using *P. aeruginosa* NEJ01R and to develop optimized culture conditions (temperature, pH and agitation) in a defined culture medium by response surface methodology.

## 2. Materials and Methods

### 2.1. Microorganism

Bacteria were isolated from lime-cooked maize wastewater collected from a local mill situated in Pedro Escobedo, Querétaro (Mexico). Samples (1 mL) of nejayote at pH 11 (without other treatments) were plated on LB agar and incubated at 30 °C for 48 h. The composition of the LB agar was (g L^−1^) casein peptone (10), yeast extract (5), sodium chloride (0.5) and bacteriology agar (15). Single colonies with different morphologies were plated again in LB agar. Three colonies (NEJ01R, NEJR5 and NEJ03R) were selected because a green coloration was observed in plates. NEJ01R was used in this study because it showed pigment production in 24 h and growth in different substrates [16]. The strain NEJ01R was characterized using Gram stain (-) and the biochemical test API 20E (bioMérieux) as *Pseudomonas aeruginosa*. Genomic DNA was isolated from pure bacterial colonies using methodology described by [35]. A nucleotide sequence analysis (16 rRNA gene) was performed at Laboratorio Nacional de Biotecnología Agrícola, Médica y Ambiental (LANBAMA) (San Luis Potosi, México). The resulting rRNA sequences were submitted to the non-redundant nucleotide database at GenBank using the Basic Local Alignment Search Tool (BLAST) program to determine its identity. *P. aeruginosa* NEJ01R sequence showed 92.30% similarity to *P. aeruginosa* ACR20 with the NCBI Accession Number CP058333.1. The strain was stored on Luria-Bertani agar (LB) at 4 °C.

### 2.2. Statistical Design and Optimization by Response Surface

To optimize conditions of culture for biomass production and PYO generation in the defined culture medium, a central compound design was chosen with three experimental factors: temperature, pH and agitation speed. Design analysis and subsequent analysis of the experimental data were recorded in Statgraphics Technologies, Inc. (The Plains, Virginia, USA, version 15.0). Appendix A shows the design matrix. Experimental factors evaluated were obtained from preliminary experimental results. Nineteen experiments were achieved because five central points were selected to complete the lack-of-fit test. Biomass and PYO were the response variables and were measured randomly at 48 h, by duplication. The following equation shows the number of experiments achieved:(1)$$N=2^{k}+2k+cp$$
where *N* is the number of experiments, *k* is the number of experimental factors and *cp* denotes central points. To assure the rotatability and orthogonality in the experimental design, α was chosen with the next equation:(2)$$\alpha ={\left(\frac{{\left(FxN\right)}^{1/2}-F}{2}\right)}^{1/2}$$
where *F* = 2*k*. An analysis of variance (ANOVA) was performed to inspect the response surface model. The fitted polynomial equation was then expressed in the form of three-dimensional response surface plots to show the relationship among the responses and the experimental levels of each independent variable.

### 2.3. Culture Conditions

The inoculum was obtained from a liquid culture using LB broth. The composition of the LB broth was (g L^−1^) casein peptone (10), yeast extract (5) and sodium chloride (0.5). The liquid culture (50 mL in 250 mL Erlenmeyer flasks) was inoculated with a single fresh colony of *P. aeruginosa* NEJ01R (LB agar, 30 °C, for 24 h). The inoculated LB broth was incubated at 150 rpm, 30 °C, for 24 h.

All experimental units (EUs) generated by statistical design were made in 250 mL Erlenmeyer flasks with 50 mL of LB broth (defined medium, LB) and sterilized in an autoclave at 121 °C for 15 min. After this, each EU was inoculated with 1 mL of an inoculum culture. The EUs were incubated for 48 h using different combinations of conditions (pH, temperature and agitation) established by the statistical design.

### 2.4. Biomass Determination

The biomass was determined by the dry weight method. Aluminum dishes were pre-dried to constant weight in a drying oven (100 °C, for 1 h). Later, they were allowed to cool in a desiccator, and then their initial weight was recorded. After 48 h of incubation, cultures were centrifuged at 6600× *g* for 10 min. The supernatant was decanted, and the formed pellets were transferred to an aluminum dish and were placed in a drying oven at 100 °C for 1 h. Finally, the dishes were placed in a desiccator, and their final weight was measured. The biomass concentration was determined by the weight difference and reported in g L^−1^.

### 2.5. PYO Determination

The decanted supernatants were used to quantify PYO by a liquid–liquid extraction method. Equal volumes of supernatant and chloroform (3 mL) were added and vigorously vortexed. The organic phase (blue) was separated, and a similar volume of 0.2 N HCl was added. The mix was vigorously stirred in a vortex, and the aqueous phase (red) was separated to measure it in a UV-Vis spectrophotometer (Genesys™ 10S, Thermo Scientific, WI, USA) at 520 nm. The absorbance obtained was multiplied by the factor 17.1 to obtain μg mL^−1^ of PYO [5].

### 2.6. Microbial Growth Kinetics and PYO Production

The kinetics of microbial growth were determined in cultures of *P. aeruginosa* NEJ01R under conditions optimized for growth and PYO production according to surface response. Samples were taken every two hours for 48 h of culture incubation. Serial dilutions (up to 10^−8^) were made in 0.1 mM phosphate buffer (pH 7.0) for each sample. One milliliter of the dilutions was added on Petri dishes with agar standard methods and incubated (30 °C). After 24 h, a bacterial count (CFU mL^−1^) was performed. In addition to bacterial count (CFU mL^−1^), the determination of PYO was made following the methodology described in Section 2.5 [5].

### 2.7. Valorization of Maize Wastewater

The wastewater was used as a culture medium (raw nejayote, RN) without treatment under the conditions (composition and pH) used in a traditional process. Afterward, RN was processed using the methodology reported by [25] employing a flocculating agent and pH adjustment (7.2). The fraction obtained was labeled as clarified fraction (CL). The CL fraction was filtered using a Pellicon^®^ system (30 kDa Biomax Mini polyethersulfone cassette, Merck Millipore, MA, USA). After this ultrafiltration process, the CL fraction was divided into retentate fraction (RE) and filtered fraction (FI). Fifty milliliters of RN and each fraction (CL, RE and FI) in 250 mL Erlenmeyer flasks was inoculated with 1 mL of an inoculum culture of *P. aeruginosa* NEJ01R. Submerged fermentations were incubated for 48 h using operational conditions (pH, temperature and agitation) established as optimal by the statistical design. All experiments were performed in triplicate.

### 2.8. Characterization of Maize Wastewater and Its By-Products

Before and after submerged fermentations with *P. aeruginosa* NEJ01R in optimized conditions (statistical design), RN and fractions were characterized. The concentration of total solids in the RN and three fractions (CL, RE and FI) was determined. Porcelain capsules (pre-dried to constant weight) were used in a drying oven at 100 °C for 1 h, and their initial weight was measured. One milliliter of the sample was placed and dried at 100 °C for 1 h and allowed to cool to room temperature, and the final weight was recorded. The concentration (g L^−1^) of total solids was determined by the weight difference method. The concentration of insoluble solids was determined using porcelain Gooch crucibles and Whatman No. 1 filter; both materials were pre-dried to constant weight for 1 h in a drying oven. Subsequently, 2 mL of each sample was filtered in porcelain Gooch crucibles, and the crucibles with the filter paper were dried in an oven (100 °C, 1 h). Finally, their final weight was measured. The concentration of insoluble solids was determined by the weight difference method. The soluble solids were obtained by the difference between the weight of total solids and the weight of insoluble solids. The ashes in the fractions were also determined, and porcelain crucibles were pre-dried to constant weight in a drying oven (100 °C, 1 h). Then, 1 mL of the sample was placed in crucibles until dry on a heating plate (100 °C). The dry crucibles with samples were placed in a muffle (550 °C, 2 h). Finally, the final weight was measured. The ash concentration was determined by the weight difference method and reported in g L^−1^.

### 2.9. Ferulic Acid Determination

The analysis of ferulic acid was carried out by an H-Class Acquity ultra-performance liquid chromatography (UPLC) system (Waters^®^, Mildford, MA, USA) using a C18 Waters UPLC BEH C18 column (50 mm × 2.1 mm i.d., 1.7 µm). Water/acetonitrile as a mobile phase was at 90:10 ratio, using a flow of 0.3 mL min^−1^, maintaining the temperature of the column at 30 °C, with an injection volume of 5 μL and elution times of 5 min. The UPLC system was coupled to the quaternary pump, refrigerated autosampler and an extended wavelength photodiode array detector (PDA detector, Waters^®^, Mildford, MA, USA). In PDA detection, the system was employed by recording a wavelength of 320 nm. The chromatographic data were obtained and processed by the software Empower3 (Waters^®^, Mildford, MA, USA).

### 2.10. PYO Identification

An analysis was performed on a UPLC system with a cooling autosampler, a quaternary solvent manager, an oven for an analytical column, a PDA Detector and an Acquity QDa mass detector (Waters^®^, Mildford, MA, USA). The QDa mass detector is a compact, single, quadrupole mass detector equipped with an electrospray ionization (ESI) interface. A Waters UPLC BEH C18 column (50 mm × 2.1 mm i.d., 1.7 µm) was used at 30 °C. Water (HPLC grade) with 0.1% formic acid was used as mobile phase A, and mobile phase B was acetonitrile. The workflow was 0.3 mL min^−1^, with a mobile phase A:B ratio of 95:5 *v*/*v*. The injection volume was 10 µL, with the autosampler kept at 15 °C. For mass detection, the QDa detector was operated in an electrospray positive-ion mode and the cone voltage was set at 10 V. The desolvation temperature was set at 600 °C. The Mass Spectrometry (MS) scan mode was used for a full mass spectrum between *m*/*z* 100 and 300, acquired with a sample rate of 5 points/s. In PDA detection, the system was employed by recording a multiwavelength set in the wavelength range of 210–800 nm. PYO (Sigma-Aldrich, St Louis, MO, USA) was used as the standard. All samples and standards were filtered by a 0.2 µm nylon membrane. Areas of peaks were determined using the Empower3 chromatography software (Waters^®^, Mildford, MA, USA).

### 2.11. Electrochemical Study

Cyclic voltammetry (CV) was performed at different scan rates on a conventional three-electrode cell employing a glassy carbon electrode as the working electrode, a Pt electrode as the counter electrode and Ag/AgCl as the reference electrode (3 M NaCl). Measurements were carried on a potentiostatic/galvanostatic device (Biologic VSP, Grenoble, France). Ten milliliters of sodium phosphate buffer (0.2 M, pH = 7) was used as the supporting electrolyte in the presence of PYO (0.14 mM) extracted from *P. aeruginosa* NEJ01R. Prior to each measurement, the solutions were purged with highly purified nitrogen for 10 min, and the compensation of the ohmic drop was carried out using electrochemical impedance spectroscopy. The resistance to the solution was measured at a frequency of 100 kHz with a sinus amplitude of 10 mV. The potential was established in its open-circuit value. The ohmic drop was set at 85%.

## 3. Results and Discussion

### 3.1. Culture Condition Optimization in Defined Medium

In order to determine the interaction of different variables (factors) with microbial growth and metabolite production by *P. aeruginosa* NEJ01R, a design experiment coupled to surface response methodology was employed. A central composite design also allowed us to evaluate the optimal conditions in the biomass and PYO produced in a defined culture medium (LB broth). At present, there are no studies that report optimization using statistical tools to determine the best combination of factors to obtain high concentrations of PYO in biological processes using LB broth. Table 1 shows the total experiments and results of biomass and PYO obtained.

Analysis of results was done considering the variability from experimental factors and total error (Table 2). Experimental conditions evaluated in this work allowed us to have several values of biomass and PYO. In some experiments, the biomass was preferred instead of PYO. Maximum biomass and PYO generated were 0.974 g L^−1^ and 2.82 µg mL^−1^, respectively; and minimum values were: biomass = 0.02 g L^−1^ and PYO = 0.0001 µg mL^−1^.

Moreover, the factors’ levels were adequately separated to estimate the effects on the responses. Coefficients of regression equation are displayed in Table 3 considering the following equation:(3)$$R=\beta_{0}+\beta_{1}X_{1}+\beta_{2}X_{2}+\beta_{3}X_{3}+\beta_{4}X_{1}^{2}+\beta_{5}X_{1}X_{2}\;+\;\beta_{6}X_{1}X_{3}+\beta_{7}X_{2}^{2}+\beta_{8}X_{2}X_{3}+\beta_{9}X_{3}^{2}$$
where *R* is the estimated value of biomass or PYO and *β* is the polynomial coefficient.

The reduced model was achieved considering the factors with significant effect for biomass and PYO. Pareto charts show only the factors and interactions with effects in the responses (Figure 1). For biomass generation, the pH and the pH/pH interaction had a negative effect, while the interaction agitation/agitation was positive. Moreover, the last interaction had a higher value than pH and pH/pH interaction obtaining a high biomass value, indicating that moderate agitation and low pH values promoted an increase in the biomass. For the case of PYO, agitation and temperature/temperature and temperature/pH interaction had a negative effect on PYO generation. In the same case for biomass, to have a higher production of PYO, temperature, pH and agitation values should be at low levels (Table 1). Response surface plots (Figure 2) were constructed from the regression equations for biomass and PYO generation.

Response surface plots (Figure 2) confirmed the principal effect of temperature, pH and agitation rate. In the case of biomass, maximum values were obtained using low temperature, low pH and low agitation values. PYO response surfaces had a different behavior because low–medium temperatures also increased PYO generation. pH values maintained the PYO in a medium value, and the effect of agitation were similar to that of pH. Simultaneous analysis was done to obtain the maximum generation of biomass and PYO considering the temperature, pH and velocity of agitation. In this case, the desirability as well as factor conditions are shown in Table 3. The maximum biomass and PYO generation calculated from the lineal reduced regression equation are also shown. Simultaneous analysis showed that the factor values obtained the maximum biomass and PYO considering the desirability. The best conditions for PYO production obtained by the surface response methodology were used in cultures, reaching PYO = 2.05 µg mL^−1^. These conditions were evaluated to continue the kinetic production of biomass and PYO.

### 3.2. Microbial Growth Kinetics and PYO Production

Figure 3a shows the kinetics of microbial growth and PYO production of *P. aeruginosa* NEJ01R, where the lag phase lasted for 5 h and subsequently started the exponential phase until 13 h. The stationary phase stayed until 43 h, after which it finally entered the decaying phase. PYO production was not associated with growth because production started at 16 h once the stationary phase began; then, it did not exhibit the same behavior as bacterial growth. The maximum production of PYO was at 40 h, three hours before starting the death phase. This microbial growth kinetics and PYO production were carried out in a defined LB medium, which is considered as a standard for the incubation period in the culture medium and is able to establish the maximum production time of PYO. Other studies report that the maximum production is after 48 and 72 h of growth, as is the case for several strains of *P. aeruginosa* isolated from surgical samples, minced meat and infected wounds [36]. The processing time is an important variable to study for the design of processes. In this work, it was possible to obtain a maximum concentration of PYO (3.3 µg mL^−1^) in 40 h using a defined synthetic medium (LB) under optimized conditions in the statistical design. While in other strains of *P. aeruginosa*, such as KU-BI02, concentrations of 2.560 µg mL^−1^ have been produced in 72 h using King’s A medium supplemented with soybean seeds [17]; in strains *P. aeruginosa* R1 and *P. aeruginosa* U3, both in King’s A medium, 4.5 µg mL^−1^ and 2.4 µg mL^−1^ of PYO were obtained, respectively, at 96 h of incubation [5].

### 3.3. Maize Wastewater (Nejayote) as a Substrate to PYO Production

Three fractions of lime-cooked maize wastewater (nejayote) were obtained (CL, RE and FI) in addition to raw nejayote (RN) to evaluate PYO production by *P. aeruginosa* NEJ01R. Figure 3b shows results concerning the samples used separately as the sole carbon source for fermentation of *P. aeruginosa* NEJ01R and PYO production, using optimal culture conditions established in the defined medium LB. RN substrate was only able to be used as a carbon source by *P. aeruginosa* NEJ01 because only biomass production was obtained (1.2 g L^−1^). The PYO production was probably hindered due to the alkaline pH (11) of raw nejayote. These results can be compared with treatments in the experimental design, since in those that presented an alkaline pH of 11, a minimum PYO production was achieved, such as treatment 10 (Table 1). In the case of the CL fraction, values of biomass production were similar to those obtained in the RN substrate, but with PYO production. Contrary to the RN substrate, pH was adjusted to 7.2, which had a positive effect on metabolite production considering that the optimum pH to produce PYO was 6.92, according to the statistical design (Table 3).

The RE fraction had biomass production values higher than that of other fractions. RE and CL fractions were adjusted to pH 7.2. This condition was favorable for both biomass and PYO production. In the RE fraction, there were molecules larger than 30 kDa, which probably favored PYO production. Ultrafiltration (UF) membranes from 3 to 30 kDa concentrate high-molecular-weight components such as proteins, hydrolysates and phenolic fractions according to [37]. Nejayote contains polysaccharides such as feruloylated arabinoxylans, which are non-starchy compounds derived from endosperm cell walls of cereals and formed with a linear -(1→4)-xylopyranose backbone and -L-arabinofuranose residues as side chains on O3 and O2 and O3 [29]. Arabinoxylans can present some arabinose residues ester-linked on (O)-5 to ferulic acid (3-methoxy-4-hydroxycinnamic acid). These compounds display ferulic acid concentrations of 0.6 μg L^−1^, as well as an arabinose-to-xylose ratio (A/X) of 0.57–0.65 with a molecular weight of 60 kDa [26]. In this work, the membrane used for the separation of fractions was 30 kDa, so it was possible that feruloylated arabinoxylans were in the RE fraction according to [25,27,28]. In that case, *P. aeruginosa* NEJ01R should have an appropriate enzymatic machinery to use this type of polysaccharide as a carbon source, such as polysaccharide-degrading enzymes. Therefore, it is feasible that *P. aeruginosa* NEJ01R produces enzymes to degrade the high volume of polysaccharides present in nejayote RE fraction with a positive effect on PYO production. Different strains of *P. aeruginosa* have demonstrated biosynthesis capacity for effluents of biotechnological processes (with complex composition) such as biodiesel production [16]; raw substrates, such as cottonseed meal, grape seeds, pea pods, taro leaves and olive wastes, hydrolyzed by acids [5]; and others, such as ground corn kernels, ground soybean seeds, potato cooking water, ground watermelon seeds and groundnut [17]. Different concentrations of PYO can be obtained in a defined medium such as nutrient broth and King’s A supplemented with 1% of different raw substrates in submerged fermentations with *P. aeruginosa* KU-BI02. Ground corn kernel produces 0.3414 μg mL^−1^ of PYO when added to nutrient broth, while adding the same amount to King’s A medium yields 1.877 μg mL^−1^ of PYO. Using the same base medium (nutrient broth), the addition of 1% ground soybean also produces 0.1702 μg mL^−1^, which increases to 1.702 μg mL^−1^ when sweet potato cooking water is added. Similarly, minimum concentrations of PYO (0.5106 μg mL^−1^) are obtained using King’s A medium supplemented with ground watermelon seeds until reaching 2.560 μg mL^−1^ of PYO when the same King’s A medium is supplemented with ground soybean [17].

On the other hand, the FI fraction displayed a lower production of biomass and PYO, which indicated that its components had a negative effect on its production. This fraction contains low-molecular-weight polysaccharides (molecules smaller than 30 kDa) and free hydroxycinnamic acids according to [25,27,28]. In nejayote, ferulic acid, *p*-coumaric acid and other oligomers have been reported [21,26], as well as antimicrobial capacity. *p*-Coumaric acid inhibited the growth of *Escherichia coli* (99.9%) at 1000 μg mL^−1^, while *Staphylococcus aureus* and *Bacillus cereus* were inhibited at 500 μg mL^−1^ [38]. Other reports indicate that 0.1 g L^−1^ of ferulic acid is the minimum inhibitory concentration for *E. coli* and *P. aeruginosa*, and 1.1 g L^−1^ and 1.25 g L^−1^ for *S. aureus* and *Listeria monocytogenes*, respectively [39]. The highest proportion of hydroxycinnamic acids present in maize wastewater are linked to other molecules such as carbohydrates. However, a significant amount of hydroxycinnamic acid has also been found in free form. The concentration of ferulic acid in nejayote depends on the nixtamalization condition process and corn variety. In nixtamalization processes using different types of maize, ferulic acid concentration has been reported with values of 99.1 for white corn, 96.74 for yellow corn, 88.63 for red corn and 84.95 mg/100 g of dry matter for blue corn [23]. In this work, for the FI fraction, ferulic acid concentration was 2.1 g L^−1^, a higher concentration than that reported previously. The lowest concentration of PYO obtained in the nejayote fraction (FI) was 0.67 μg L^−1^. These results indicated that *P. aeruginosa* NEJ01R strain displayed resistance to the antimicrobial effect of ferulic acid and inhibition of PYO production.

In nixtamalization wastewater fractions, a similar ferulic acid concentration was found, which prevented an antimicrobial effect as shown in Figure 3b. The effect of hydroxycinnamic acids on PYO production by *P. aeruginosa* was reported previously, demonstrating that 4 mM of ferulic acid can inhibit about 20% of PYO production [40]. In another study, it was shown that phenolic compounds, such as methyl gallate, have a negative effect on PYO production related to quorum sensing system and other molecules that inhibited the formation of biofilm, motility, proteolytic, elastase or rhamnolipid production in *P. aeruginosa* PAO1 [41]. In our results, after treatment with *P. aeruginosa* NEJ01R, the initial concentration of ferulic acid decreased by 10% for CL and FI fractions. In the RE fraction and RN substrate, values decreased to 15% and 20%, respectively, suggesting that *P. aeruginosa* NEJ01R can use molecules of ferulic acid as a carbon source.

Furthermore, during the cooking process of corn, to obtain nixtamal, 0.5–2% of calcium hydroxide is added, so this compound may be present in the wastewater obtained. It has been reported that the concentration of this compound in nejayote can be up to 1526.21 mg L^−1^ [22]. The calcium hydroxide residues present in nixtamalization wastewater can have a negative impact on low PYO production using the FI fraction since it is likely that a high concentration of calcium was present. It was also demonstrated that a fraction with a concentration of 3155.3 ± 5.24 mg L^−1^ of calcium was obtained using clarified nejayote filtered by a 1 kDa membrane [42]. Therefore, according to our results, the FI fraction may have had a high calcium concentration that interfered with PYO production by *P. aeruginosa* NEJ01R. In a study about the effect of calcium chloride concentration on PYO production by *P. aeruginosa* NRRL B-771 and a PaJC mutant, it was demonstrated that 50 mM of calcium chloride in the presence of 1% sucrose produced a PYO concentration of 3.53 μg mL^−1^ and 3.06 μg mL^−1^ for *P. aeruginosa* NRRL B-771 and the PaJC mutant, respectively. When the calcium chloride concentration was increased to 250 mM, the PYO production decreased to 1.66 μg mL^−1^ and 2.51 μg mL^−1^ for *P. aeruginosa* NRRL B-771 and the PaJC mutant, respectively [43]. This suggests that the concentration of calcium in the culture medium has a negative effect on the PYO production of *P. aeruginosa* NEJ01R. If we consider the membrane size used to obtain the fractions RE and FI, it is likely that a higher concentration of calcium accumulates in the FI fraction, and this would have a negative effect on PYO production related to lower concentration in the FI fraction compared with the RE fraction. Then, the ultrafiltration process utilized showed an efficient system to fractionate nixtamalization wastewater that is feasible in PYO production by *P. aeruginosa* NEJ01R in order to valorize maize by-products.

### 3.4. Characterization of Maize Wastewater Fractions

Figure 4 shows values of parameters evaluated for characterization of nejayote fractions before and after the treatment. The results indicated that total solid (TS) content in the RE fraction increased due to ultrafiltration (UF) treatment; it was possible to obtain a fraction concentrate with 34% more solids than that contained in the RN substrate. The composition of nejayote fractions separated by UF (100 kDa) is mainly ferulated arabinoxylans, which are phenolic compounds linked to long-chain carbohydrates [25,27]. The concentration of solids decreased by 20% after treatment with *P. aeruginosa* NEJ01R, with the highest percentage of total solids being removed, as well as the CL fraction, in which 20% of TS was also removed. Similarly, for soluble solids (SS) in the treatment of RE fraction, a higher concentration of SS (20%) was removed, and for the CL fraction, 18% was removed. The highest percentage of removal of insoluble solids (IS) was obtained in RN (50%). The crude nejayote did not receive any previous treatment as the fractions that displayed a higher initial content of IS. For CL and RE fractions, the removal percentages of IS were 22% and 29%, respectively.

The FI fraction did not present IS removal, besides being the fraction that initially displayed the lowest IS concentration and was not used as a carbon source by *P. aeruginosa* NEJ01R. However, the FI fraction presented an ash removal of 45%, and the largest ash removal was in the RE fraction with 60%, followed by the CL fraction with 34% ash removal. These results showed that treatment with *P. aeruginosa* NEJ01R achieved a greater removal of solids in the RE fraction, which indicated that the concentration of solids favored the consumption of this as a carbon source by the strain used (*P. aeruginosa* NEJ01R). On the other hand, the fraction with the lowest solid removal was the FI fraction, which was also related to lower production of biomass and PYO, demonstrating that the type of molecules present in this fraction could have a negative effect, mainly in PYO production.

### 3.5. PYO Identification

The ultra-performance liquid chromatography system (UPLC system) was employed to analyze the sample extracted and compare it with a commercial standard (Sigma-Aldrich, St Louis, MO, USA). The sample extracted from the supernatant of the culture using *P. aeruginosa* NEJ01R showed two maximum peaks of 278 and 387 nm, similar to the standard commercial product (Appendix A). The UPLC chromatogram showed a single peak with a retention time of 1.45 min for PYO standard, which was comparable to that of the extracted sample, which suggested that it was the same molecule. On the other hand, the maximum at 278 nm of PYO was also reported for a culture of *P. aeruginosa* TBH2, an extracted sample obtained with chloroform and acidified with 1M HCl, and in the same way for PYO extracted from cultures of *P. aeruginosa* N11, *P. aeruginosa* D23 and a *P. aeruginosa* clinical isolate [3,44]. Appendix A displays the mass spectra of extract sample and standard PYO, giving a result of *m*/*z* 211 (M + H), which coincides with PYO mass of 210 g mol^−1^. This result corresponds with that reported for a culture sample extract of *P. aeruginosa* BRp3 strain [45]. Our results demonstrated that the molecule extracted from the culture of *P. aeruginosa* NEJ01R was PYO.

### 3.6. Electrochemical Analysis of PYO

The redox behavior of PYO was analyzed with cyclic voltammetry. Figure 5a shows the electrochemical response of a PYO solution (0.14 mM) in the presence of 0.2 M phosphate buffer solution at pH 7. The analysis indicated a reduction process *I_c_* with a peak potential *E_pc_* of −0.269 V vs. Ag|AgCl 3 M NaCl value at 100 mV s^−1^. In the reversal scan, a reduction process *I_a_* was observed with a peak potential *E_pa_* of −0.234 V vs. Ag|AgCl 3 M NaCl at the same scan rate.

The redox behavior of PYO was studied at different scanning rates. A normalization current analysis was carried out, coming from the following equation:(4)$$\frac{i}{nFAC_{O}^{*}D_{O}^{1/2}{\left(\frac{nF}{RT}\right)}^{1/2}v^{1/2}}=\pi^{1/2}x\left(\sigma t\right)$$
where *n* is the total number of electrons in the redox reaction, *D_o_*^1/2^ is the PYO diffusion coefficient, *A* is the electroactive area, *C_o_* (PYO concentration) values are constants and *π*^1/2^
*χ(**σt)* is a dimensionless number.

This analysis suggested that the cathodic signal *I_c_* did not have diffusional complications or coupled chemical reactions (Figure 5a). On the other hand, for signal *I_a_*, the normalization of current values increased with *v*^1/2^. The anodic peak potential *E_pa_* varied significantly as the scan rate increased with shifts toward positive values. On the other hand, minimal changes in *E_pc_* values were observed when *v*^1/2^ was increased. This behavior suggested a reversible redox behavior with a possible adsorption for *I_a_*. To confirm this idea, an analysis of the peak current *i_p_* as a function of *v*^1/2^ was also performed. According to the following equation (Randles Sevcik) [46], the value of *i_p_* must present a linear relationship with *v*^1/2^.
(5)$$i_{p}=\left(2.69\times 10^{5}\right)n^{3/2}AD_{O}{}^{1/2}C_{O}^{*}v^{1/2}$$

Figure 5b presents the linear behavior of *i_pc_* with respect to *v*^1/2^, indicating that the electrochemical process *Ic* of PYO was controlled by diffusion. A change in linearity of *i_pa_* vs *v*^1/2^ was observed at high scan rates (Figure 5c). Hence, the ratio *i_pa_/i_pc_* at different scan rates was evaluated (Figure 5d). The behavior confirmed that the reversal oxidation process *I_a_* was controlled by diffusion, with adsorption of PYO significantly reduced over the electrode surface [10,47]. Finally, considering low scan rates, where the adsorption process was not presented, a value of |Δ*E_p_*| = 0.035 V was calculated, which indicated a two-electron transfer, according to Equation (6). A value of *E*_1/2_ = −0.251 V vs. Ag|AgCl 3 M NaCl, related to the redox pair of PYO, was calculated according to Equation (7) [48,49]. Different applications of PYO have been described, in agriculture, medicine and the textile industry. However, the observation in two areas of application for this molecule is limited, namely in the extracellular transport of electrons in bioelectrochemical systems, specifically as biosensors, and in microbial electrochemical technologies. The phenomenon observed at high scanning speeds due to complications in the electrode related to absorption processes could be analyzed in future research to advance the application in the mentioned areas.
(6)$$\left|E_{pa}-E_{pc}\right|=\frac{59\;\mathrm{mV}}{n}$$
(7)$$\frac{E_{pa}+E_{pc}}{2}=E_{1/2}$$

## 4. Conclusions

In this study, a coupled methodology of design of experiments and response surfaces was used to determine the optimal process conditions for obtaining PYO in submerged fermentation by *P. aeruginosa* NEJ01R. These conditions (29.6 °C, 223.7 rpm, pH 6.92) were evaluated experimentally, in LB medium, reaching values (2.05 µg mL^−1^) close to those predicted statistically (2.21 µg mL^−1^). We demonstrated that the by-product of the maize industry can be a culture medium, without supplementation, that supports the production of PYO, only in its pretreated fractions and separated by an ultrafiltration membrane. The highest PYO concentration (3.25 μg mL^−1^) was obtained in the RE fraction at 48 h. Furthermore, the high concentration of ferulic acid (2.1 g L^−1^) in the FI fraction did not have a negative effect on the growth of *P. aeruginosa*, which is related to resistance to its antimicrobial effect, although it did show a negative effect on the production of PYO. This is the first work that reports the use of wastewater as the only carbon source to synthesize PYO by a native strain of *P. aeruginosa* (NEJ01R). The molecule obtained showed redox behavior without diffusion complications at low scan rates.

Response surface plots indicated the relevance of factors with a significant effect for biomass and PYO production, indicating a positive effect using values in the low level (temperature, 21.5 °C; pH, 5; agitation, 76 rpm) for higher production in both responses. Optimized variables for PYO production in alkaline residual liquid of the maize industry and a detailed characterization establishing the composition of nejayote fractions and compounds that are involved in the metabolic induction of PYO are in progress.

## Figures and Tables

**Figure 1 microorganisms-08-01559-f001:**
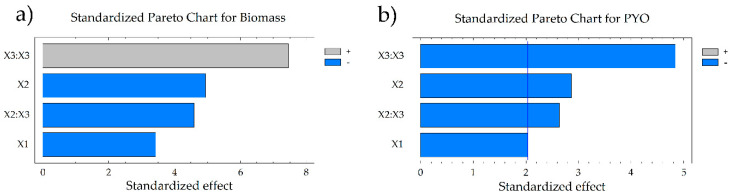
Pareto chart for (**a**) biomass and (**b**) pyocyanin (PYO) generation in the defined culture medium. The graphics show only significant effects. X1, temperature; X2, pH; X3, agitation.

**Figure 2 microorganisms-08-01559-f002:**
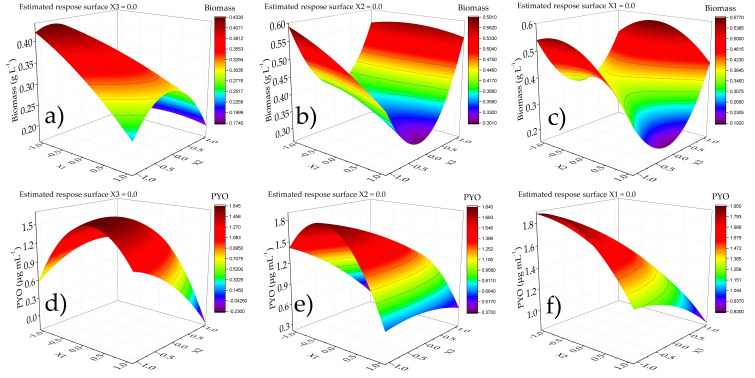
Response surface plots for biomass and PYO production estimated from regression equation in statistical analysis. (**a**) Interaction between temperature and pH; (**b**) interaction between temperature and agitation rate; (**c**) interaction between pH and agitation rate; (**d**) interaction between temperature and pH; (**e**) interaction between temperature and agitation rate; (**f**) interaction between pH and agitation rate.

**Figure 3 microorganisms-08-01559-f003:**
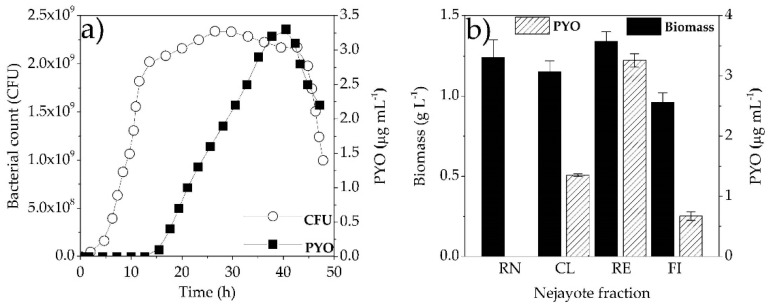
(**a**) Microbial growth kinetics and PYO production of *Pseudomonas aeruginosa* NEJ01R under optimized growth conditions; (**b**) biomass and PYO production of *P. aeruginosa* NEJ01R using raw nejayote (RN) and fractions (clarified (CL), retentate (RE) and filtered (FI) fractions) under optimized conditions.

**Figure 4 microorganisms-08-01559-f004:**
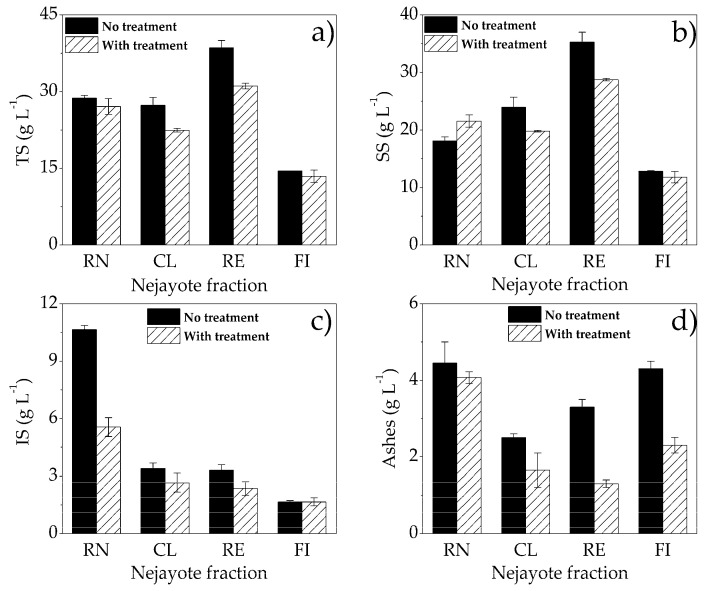
Determination of (**a**) total solids (ST), (**b**) soluble solids (SS), (**c**) insoluble solids (IS) and (**d**) ashes in nixtamalization wastewater crude (RN) and fractions (CL, RE and FI).

**Figure 5 microorganisms-08-01559-f005:**
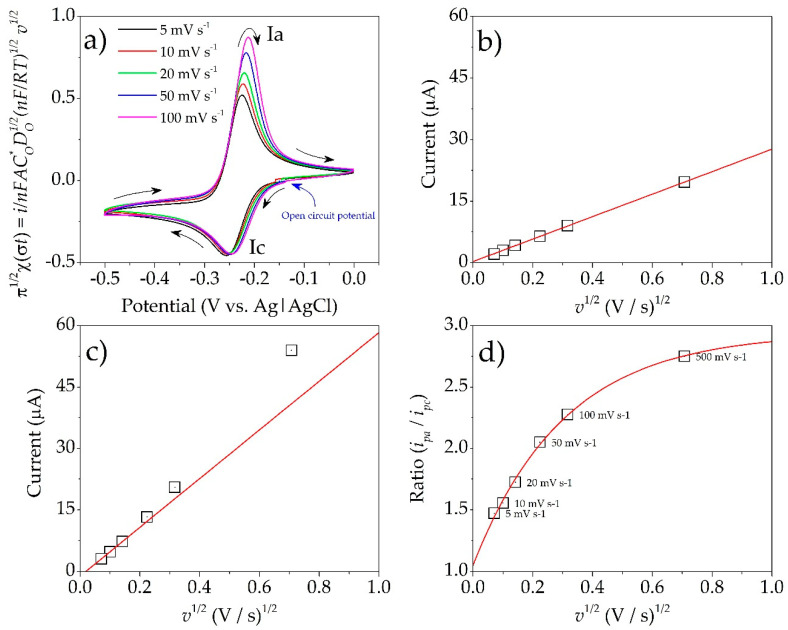
(**a**) Cyclic voltammetry of PYO (0.14 M) in current normalized at different scan rates (*v*). (**b**) *i_pc_* values for PYO (0.14 M) as a function of *v*^1/2^. (**c**) *i_pa_* values for PYO (0.14 M) as a function of *v*^1/2^. (**d**) *i_pa_/i_pc_* values of PYO (0.2 mM) as function of *v*^1/2^.

**Table 1 microorganisms-08-01559-t001:** Experimental central composite design matrix; biomass and pyocyanin (PYO) experimental results.

Unit	Experimental Code Factors			Results			
	Temperature (X_1_)	pH (X_2_)	Agitation (X_3_)	Biomass g L^−^^1^		PYO µg mL^−1^	
1	0	0	0	0.368	0.380	1.5903	1.4260
2	−1	−1	1	0.512	0.484	0.0001	0.0001
3	1	−1	−1	0.480	0.330	0.6600	0.5800
4	1	1	−1	0.318	0.278	0.4788	0.3420
5	0	0	−1.47119	0.600	0.710	2.8200	2.7800
6	0	0	0	0.388	0.380	1.5732	1.4312
7	0	0	0	0.368	0.366	1.6416	1.6758
8	−1.47119	0	0	0.440	0.470	0.5814	0.4275
9	1	−1	1	0.440	0.370	1.1900	1.2800
10	0	1.47119	0	0.036	0.020	0.0500	0.0400
11	1.47119	0	0	0.252	0.188	0.0342	0.0223
12	0	0	0	0.340	0.380	1.5390	1.4483
13	0	0	0	0.358	0.346	1.5323	1.4141
14	−1	1	1	0.320	0.398	1.4706	1.5561
15	−1	−1	−1	0.705	0.386	0.9498	1.0602
16	0	0	1.47119	0.974	0.484	0.6669	0.7011
17	1	1	1	0.370	0.240	0.1300	0.2200
18	−1	1	−1	0.364	0.356	1.4193	1.2996
19	0	−1.47119	0	0.380	0.342	2.3127	2.3598

**Table 2 microorganisms-08-01559-t002:** ANOVA for biomass and PYO generated from the defined culture medium.

Source	Biomass		PYO	
	Coefficient	*p*-Value	Coefficient	*p*-Value
Constant (X_0_)	0.3674		1.5832	
Temperature (X_1_)	−0.0390	0.0732	−0.1539	0.01757
pH (X_2_)	−0.0655	0.0042	−0.2053	0.0744
Agitation (X_3_)	0.0227	0.2869	−0.2712	0.0210
Temperature:temperature (X_1_^2^)	−0.0138	0.5700	−0.6030	0.0001
Temperature:pH (X_1_X_2_)	0.0413	0.1236	−0.3620	0.0138
Temperature:agitation (X_1_X_3_)	0.0336	0.2074	0.1838	0.1919
pH:pH (X_2_^2^)	−0.0799	0.0026	−0.1760	0.1769
pH:agitation (X_2_X_3_)	0.0333	0.2107	0.0686	0.6215
Agitation:agitation (X_3_^2^)	0.1499	0.000	0.0787	0.5405

**Table 3 microorganisms-08-01559-t003:** Calculated conditions for the maximum biomass and PYO production in the defined culture medium.

Response	Biomass Calculated, 0.71 g L^−1^ PYO Calculated, 2.21 µg mL^−1^ (Desirability = 0.9084)	
Factor	Code	Real Value
Temperature (°C)	−0.3220	29.6
pH	−0.5007	6.92
Agitation (rpm)	1.4711	223.7

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
