# Peer review of "Optimized Production of a Redox Metabolite (pyocyanin) by Pseudomonas aeruginosa NEJ01R Using a Maize By-Product"

_microorganisms, 2020, doi:10.3390/microorganisms8101559_

Round 1
Reviewer 1 Report
The manuscript “Optimized production of a redox metabolite (pyocyanin) by Pseudomonas aeruginosa NEJ01R using a maize by-product” the optimal conditions for PYO production were studied and used to obtain this metabolite in fractions of nixtamal industry wastewater. I think that sounds good and that some researchers in the field may be interested to take a look on. However, the manuscript shows some minor faults, which should be addressed (see Specific Comments).
SPECIFIC COMMENTS
Page and line numbers refer to the original PDF file, downloaded from the journal's website.
- Page 2, line 64: www.sigmaaldrich.com - Is the right form of website citing in the text? Pls, check.
- Page 3, line 135: 7,500 rpm should be converted to g in order to become independent of the rotor and centrifuge used.
- Page 7, Fig 1. The images should be resized. Image a) is very large and b) is very small and cannot be read.
- Page 10, line 347: P. aeruginosa instead of Pseudomonas aeruginosa
Reviewer 2 Report
The study is really interesting and well structured. However, what is not strongly evident is the real and effective contribution and innovation of this study in the reference context. Clearly, the scientific value of the study is undoubted, but in my opinion it should be accentuated. The language requires deep revision throughout the manuscript. Sentences are often poorly constructed and appear to be lists of steps and results
Here some suggestions:Lines 47-48. I suggest to replace this with 'a bacterial strategy used to'
Lines 56-57. Please adjust the sentence, it is not clear.
Line 68. 'to reduce their polluting capacity' maybe is better to exress as 'reduce the polluting and harmful potential'
In the introduction section are present a lot of references about previously studies, with obtained results from other researchers. I suggest to move these considerations in the discussion, in order to provide also a more critical analysis of your results, also through comparison. Instead, it would be preferable to put here consideration about the topic in general (partially provided), the main aim of the study and the originality respect to previous studies.
Line 126. Maybe is better 'for 24 h'.
Lines 133-146. The sentences are too long. Please break it into shorter sentence. Describe each passage properly.
Line 166. Probably is better 'measured' instead of 'recorded'
Lines 218-222. 'to evaluate the optimal conditions in the biomass and PYO
produced in a defined culture medium' Here you refer to the PYO yield. Please refer more properly, as you defined the optimal conditions in terms of bacterial biomass and PYO yield
Line 281: correct addition
Line 356: correct shows
Line 429. correct These
After a proper revision of language in all manuscript and minor revisions, I suggest the publication.
Author Response
Please, see the attachment.

Reviewer 3 Report
The presented results seem interesting and are important for the development of green chemistry. The work contains many interesting results. Unfortunately, the way the results are presented and the context in which they are presented require significant changes and redrafting.
The introduction is not properly adapted to the experimental work presented in the publication or the argumentation or conclusions drawn based on cited literature are not appropriate e.g.:
- There are several published works on properties and utilization of nejayote. The description of nejayote is poor.
- The cost according to sigmaaldrich is somehow surreptitious advertising and is not the best choice, definitely biased.
- For the production of fine chemicals the cost of the growth media is negligible, so there is no point to look for cheap media?
- With a very low content of isolated chemicals, cleaning/downstream processing is very costly.
- From circular economy point of view biotransformation of nejayote and recovery of very small fraction (pyocyanin, around 0.3%) leaves anyway close to 100% waste. What will be the purpose of the digestate?
- Several quoted figures/values are misleading. For example, the reported methane yield from nejayote is from the second reactor in a two-stage process. There is very little methane in the gases from the first bioreactor. There is no organic substance that can provide this (84%) concentration of methane in anaerobic digestion. Oil substrates gives the highest methane concentration. Nejayote contains a lot of carbohydrates for which, according to the stoichiometry, it can be 50-60% of methane.
The isolated strain should be deposited in a registered collection in the World Federation of Culture Collections and the accession number given.
More detailed chemical composition for the used nejayote and fractions should be provided.
Figures together with legends should be self-explanatory. Legends are incomplete or misleading ex. UCF instead of CFU.
Conclusions should be more informative and based on the results. In present form there are not acceptable.
The ms should be proofread ex. space between the value (digits) and % should be removed.
Author Response
Please, see the attachment.

Reviewer 4 Report
The authors performed research on the production of pyocyanin by Pseudomonas aeruginosa NEJ01R isolated from lime cooked maize wastewater.
The authors described their results on the following topics:
- Determination of optimal conditions using an experimental design approach using 19 different experiments as input.
- growth and production of PYC on (fractions of) lime cooked maize wastewater
- HPLC, spectral and mass-spectrometry analysis
- cyclic voltammetry of PYC.
The abstract is not written in a very clear way. E.g., It is unclear what effluent RN means. In the next line condition for maximal PYC production in defined medium are described. However, these values are different from the values mentioned in Table 3. In the next line they write an isolated line mentioning that the retentate contains the highest concentration of total solids. It is unclear from the abstract what this means.
Line 26-27 are unclear to me. What was the proposal (proposed?) of adsorption? A way to purify PYC from the medium or a phenomena that happens during the cyclic voltammetry?
Introduction. This section contains to much detail on RN (line 69-82). This is not necessary for the introduction. Furthermore I miss some information on the biosynthesis of PYC. see e.g. https://www.nature.com/articles/s41598-020-58335-6 . This paper could be nice reference of recent results from another research group.
Material and Methods
2.1: very limited information on how the strain was isolated.
Results and discussion
3.3. The separation by pH adjustment, coagulation and ultrafiltration (30 kDA) are not very high resolution methods. There is too much speculation in this paragraph.
3.4 not clear what is the value of this paragraph to the main topic of the paper.
3.5 line 384 - 388 ae not very useful. This does not make the method and results by the authors more valuable.
3.6. does not add anything to the paper.
In general the paper is not very well focused. Several ideas and analytical methods have been performed without a clear line of research.
Furthermore, the price of PYC as reported by Sigma-Aldrich is far from the price that must be paid from an industrial producer in a business to business context. The price of PYC from Sigma-Aldrich is too high.
Author Response
Please, see the attachment.

Round 2
Reviewer 3 Report
The manuscript has been properly corrected. Good luck with continuing the research the authors wrote in response.
Author Response
We appreciate your revision. The comments and suggestions were very important to improve the manuscript.
Reviewer 4 Report
The authors addressed most of my comments in a satisfactory way.
However, in the new textual additions I noticed the following:
line 49: I do not understand the phrase PYO has a bacterial strategy .... PYO is a molecule and this molecule inhibits certain biological processes. A molecule does not have a strategy.
line 52: antiprotozoal activity is another feature. It does not belong to the inhibition of Candida and Aspergillus. These two organisms are not protozoa but a yeast and a fungus respectively.
line 60: .... in the secretome of Pseudomonas ....
line 64: ...carried out by direct mechanisms electron transfer .....
line 94: very low BOD value. Or is this determined in the waste water directly without proper pH adjustment? The BOD culture that was used might not be very well adapted to this substrate.
line 97 between the brackets three types are mentioned but they are all the same.
line 99: applicable is a better word than disposable.
line 117: Bacillus megaterium was able to transform ....
line 119: obtaining bioenergy or used for growth of microalgae.
line 120 .... can be exploited....
line 128: I meant here: What kind of growth media were used, what grow temperature was used. Why was this specific colony selected
line 131: You do not isolate DNA with primers
line 168: aluminum dishes were not adjusted to constant weight but predried until constant weight
line 172: supernatant was decanted
line 177: The decanted supernatants were ...
line 208, 213, 218: predried to constant weight.
line 231-232. ... "with photodiode array" is redundant. It is obvious from the name of the detector.
line 267: if there are no studies known, why does the sentence end with two references?
line 381 and in other places. 4 digit accuracy for PYO concentrations is very unlikely
Author Response
Please, see the attachment.
